behaviour/cognition/psychology

nonverbal coordination, deception, cognitive load, motion capture, mimicry

**Author for correspondence:**
Sophie Van Der Zee
e-mail: vanderzee@ese.eur.nl

# A liar and a copycat: nonverbal coordination increases with lie difficulty

Sophie Van Der Zee[1], Paul Taylor[2,3], Ruth Wong[2], John Dixon[4] and Tarek Menacere[2]

[1]Erasmus Universiteit Rotterdam, Rotterdam, Zuid-Holland, The Netherlands
[2]Lancaster University, Lancaster, UK
[3]University of Twente, Enschede, Overijssel, The Netherlands
[4]The Open University, Milton Keynes, Buckinghamshire, UK

 SVDJ, 0000-0002-3457-5916; PT, 0000-0001-8743-7667; JD, 0000-0002-3972-6437

Studies of the nonverbal correlates of deception tend to examine liars' behaviours as independent from the behaviour of the interviewer, ignoring joint action. To address this gap, experiment 1 examined the effect of telling a truth and easy, difficult and very difficult lies on nonverbal coordination. Nonverbal coordination was measured automatically by applying a dynamic time warping algorithm to motion-capture data. In experiment 2, interviewees also received instructions that influenced the attention they paid to either the nonverbal or verbal behaviour of the interviewer. Results from both experiments found that interviewer–interviewee nonverbal coordination increased with lie difficulty. This increase was not influenced by the degree to which interviewees paid attention to their nonverbal behaviour, nor by the degree of interviewer's suspicion. Our findings are consistent with the broader proposition that people rely on automated processes such as mimicry when under cognitive load.

## 1. Introduction

Studies of human deception and nonverbal behaviour typically focus on the acts of interviewees who are tasked with lying or telling the truth, or interviewers who are tasked with determining the veracity of the account. Few consider the joint nature of conversation [1,2]. This is a surprising oversight because most explanations for why behaviour changes when we deceive are rooted in the impact that an interlocutor can have on an interviewee [3,4]. For example, the anxiety associated with deception stems from the reactions of the recipient of the lie, while the increase in cognitive load associated with lying is believed to stem largely from the need to maintain a coherent account while monitoring the interviewer's reactions.

This suggests that behavioural correlates of the cognitive and social processes that underpin deceptive behaviour are probably best observed in the dynamics of interpersonal behaviour.

One behaviour that is repeatedly observed as central to both interpersonal interaction and deception is nonverbal coordination; the synchronous movements that occur between interaction partners as they tend to imitate the behaviours of one another within a short time window [5]. When partner movements are similar in form as well as timing, then the behaviour is described as nonverbal mimicry [6]. Such coordinating is likely to have evolved to allow individuals to maintain harmonious relationships with group members [7] and both coordination and mimicry have been shown to facilitate social behaviour [8–10]. However, there is a darker side. In the animal kingdom, mimicry of the behaviours of prey is a deceptive technique that allows many predators to survive [11]. In humans, an interviewer mimicking an interviewee's posture and mannerisms can increase the disparity in truthteller and liar behaviour [12], but those who mimic a deceiver may become 'blind' to the cues and are less able to recognize the other's deceit [13].

In these studies, interviewers were instructed to match the behaviour of the interviewee, but coordination can occur on both conscious and automatic levels [14]. For example, Duran & Fusaroli [15] found greater head movement coordination in deceptive conversations compared to honest conversations, and the 1/30th-second granularity of their image comparison method makes it unlikely to be the result of controlled behaviour by participants. Non-conscious coordination is particularly relevant to the deception context because liars' capacity to control their behaviour is one of the main obstacles in the search for universal cues [16]. If people are unaware of their interpersonal coordination, then changes in that behaviour may provide a useful measure for testing hypotheses regarding the social and cognitive changes that accompany lying.

Arguably, lying may affect coordination in two opposing ways. One way is to increase coordination as the cognitive demands of deception leave the liar with less resources to control their social behaviour. Studies of brain activity and motor synchronization [17,18] suggest that leading one's own rhythmic behaviour, rather than following another's behaviour, requires inhibiting the representation of the other's actions (or at least enhanced representation of one's owns actions), and thus increased cognitive effort. While mimicry can be subject to a top-down response modulation that enables its use for social advantage [19], this strategic form is reliant on executive control, which has been shown to diminish during cognitively demanding tasks. For example, Van Leeuwen et al. [20] found their participants responded quicker to the finger movement of an actor compared to a spatial cue, but only when they were cognitively loaded by a secondary task. Lying is one such task that is cognitively demanding because liars must suppress their true knowledge, create a coherent false story, control their own behaviour and monitor the responses of the interviewer [21–24]. Indeed, studies of indicators of cognitive load, such as pupil dilation [25] and brain activity [26], confirm the cognitively taxing nature of lying. Thus, if liars have less cognitive resources to dedicate to managing their social signals, then they might be expected to demonstrate heightened coordination.

The other way in which lying may affect coordination is related to people's tendency to freeze in response to both physical stressors [27] and social threats [28]. This behavioural 'freezing' would arguably lead to decreased coordination. Cues to deceit such as increased tenseness, pupil dilation and higher pitch indicate that lying can be identified as a stressor [21] and analyses of interviewees' behaviour have shown that liars often freeze by reducing body movements [16]. Additionally, research suggests that some liars deliberately limit their movement as a way of avoiding the presentation of deceptive cues [29]. For example, Burgoon & Buller [1] found that, compared to truth-tellers, liars controlled their behaviour by becoming more formal, restrained and tense, which disrupted the typical interaction pattern. This evidence suggests that liars have the capacity to 'over-control' their behaviour [30] in a way that makes them appear rigid, rehearsed and unnatural [16,31] and inadvertently curtails spontaneous coordination. Moreover, it is conceivable that this over-controlled behaviour may increase an interviewer's suspicion, which may in turn affect the interviewee's behaviour and disrupt the natural interaction dynamics [1]. Thus, rather than nonverbal coordination increasing as a result of liars having less cognitive resources, liars who freeze might be expected to show reduced coordination compared to truth-tellers. These two possibilities—that cognitive demands lead to an increase in coordination, or that stress-induced freezing will lead to a reduction in coordination—are the focus of experiment 1.

## 2. Experiment 1

To examine the impact of deception on nonverbal coordination, we compared the nonverbal behaviour of interviewers and interviewees across four statements: a truthful account, a concealed lie (i.e. hiding

cheating), a fabrication (i.e. making up a memory of an event not experienced) and a fabrication told in reverse order (i.e. the same event, but told from end to beginning). These statements were designed to be progressively more difficult for the interviewee. Lying is usually more cognitively demanding than truth-telling [21], fabrications are more demanding than concealments [30], and accounts given in reverse order are more demanding than those given in normal order [32]. Thus, the concealed lie formed our 'easy' lie condition, the fabrication our 'difficult' lie condition and the fabrication in reverse order our 'very difficult' lie condition. Given the two literatures described above, we made two competing hypotheses: that behavioural coordination between the interviewer and interviewee would increase with task difficulty (H1a) and that behavioural coordination would decrease with task difficulty (H1b).

## 2.1. Method

Experiments 1 and 2 were approved by the Lancaster University Research Ethics Committee, and is in line with the World Medical Association Declaration of Helsinki. All participants gave informed consent in writing before taking part. The data and electronic supplementary material are openly available at https://github.com/sophievanderzee/DeceptionMimicry.

### 2.1.1. Participants

Ninety-eight male students from Lancaster University (age $M = 20.9$ years, range: 18–36) acted as either an interviewee or interviewer. We recruited only male participants to avoid the impact that sex may have on coordination [33]. The interviewees ($n = 49$) participated for approximately 70 min in return for payment of £8. The interviewers participated for approximately 40 min in return for £5. Six pairs of participants were excluded because of failures in the automatic recording of nonverbal behaviour, leaving 43 pairs. Of these 43 pairs, body movement data of two pairs and right-hand movement data of 10 pairs is missing owing to sensor failure. A sample size of approximately 15 pairs per comparison was established using an effect size derived from pilot work in which 16 white-British interviewees described two experiences: an informal 'coffee shop' chat with another participant and a formal negotiation game completed against a participant. They only experienced one of these two contexts and so had to fabricate the one not experienced. An effect size from this study was derived using the same measure of nonverbal coordination used in this paper, calculated from the data of a WiTilt placed on the chest of the interviewee and a confederate interviewer. Using the effect size derived from this pilot work ($f = 0.549$; correlation among repeated measures $= 0.592$), an *a priori* repeated measure comparison using G*Power [34] with $\alpha = 0.05$ and $1\text{-}\beta = 0.95$, suggested a minimum required sample size of nine pairs per condition.

### 2.1.2. Procedure

The experiment comprised two stages: a pre-interview stage in which the interviewee completed three tasks, and an interview stage in which the interviewer questioned the interviewee about the completed tasks. At interview, all interviewees gave a truthful account and easy lie, and then half gave a difficult lie while the other half gave a very difficult lie. For these last two lies, the content was identical but its retelling was altered; half were required to give an account in forward order and half in reverse order. This mixed design was necessary to avoid any practice effects that would emerge from giving the same account twice.

*Pre-interview.* The pre-interview tasks varied in difficulty because they comprised a truth, a concealment and a fabrication in either forward or reverse order.

*Truth task.* The first task involved having an informal conversation with a confederate who was posing as a participant. The experimenter instructed participants to talk about whatever they wished, and she reassured them that their conversation was not being recorded. She then left the room and returned after 5 min.

*Easy lie task.* In the second task, the interviewee and confederate solved a wooden puzzle together in the space of 5 min. The experimenter indicated that previous participants had no trouble doing so. This last instruction sought to induce pressure on the interviewee to complete the puzzle. When leaving the room, the experimenter 'accidentally' left the solution to the puzzle in a bag on a side table, where the confederate 'discovered' it and subsequently encouraged the interviewee to cheat. Upon return, the experimenter pretended to note the instructions, and asked the interviewee and confederate whether they had used them. The confederate gave the interviewee time to admit to the cheating. If he did not do so, the confederate explained that they had used the solution together to solve the

task. In both cases, the experimenter responded: 'Sorry, that's my fault. I do not think it's really a problem for the experiment. It's just that I started my PhD here recently and my supervisor is going to teach me how to code the interview videos. Would you mind not mentioning seeing the instructions when you get asked about them in the interview?' All interviewees agreed to this concealment request.

*Difficult lie and very difficult lie task.* These two conditions relied on a single task. The 'difficult lie' and 'very difficult lie' conditions were created by asking participants at interview to give their account in either a forward (difficult lie) or reverse (very difficult lie) order (see *Interview* below). For this third task, the interviewee and confederate were separated. The experimenter 'selected' the confederate and asked the interviewee to wait in the room while she and the confederate went to another room. After a couple of minutes, the experimenter returned and told the interviewee the following: 'For this part of the task you are going to have to use your imagination by making up a story about playing a game of Cluedo (also known as Clue) with three other players. The other participant has just gone to meet these three other players but the game is a set up. The three players are confederates who will act in certain ways that move the game to a predetermined outcome. The interviewer has information about this game and his task is to try and work out whether the other participant or you were the fourth player in the game. So, your task is to convince the interviewer that you played Cluedo with the three confederates. If you do this successfully, you will be entered into a prize draw to win an iPod.'

The interviewee was given information (e.g. names and pictures of the other participants, a map of the room and the Cluedo board) and 10 min to prepare their account of playing Cluedo. Cluedo provides a rich and relevant task for participants as they can fabricate several game components that resemble forensically relevant factors—a crime, the different locations and their connections, people, weapons, and the importance of order and time to events. It has the merits of being both a bounded task, so that there is some limits to what liars fabricate, but also sufficiently varied in detail that there will probably be variety in responses. The components of the task are also commonplace to crime stories, so they have a degree of familiarity for the participant even if they are not regular Cluedo players.

Once the 10 min ended, the interviewee was brought to another room for the interview stage. In all cases, the interviewee expressed being confident enough to talk about the game. In debrief, none reported realizing that their partner during the first two tasks was a confederate.

*Interview.* Before entering the interview room, the experimenter reminded the interviewee that there would be questions on the informal conversation to which he should respond truthfully, questions on the wooden puzzle to which he should conceal that he had cheated, and questions on the Cluedo game to which he should fabricate a story and convince the interviewer that he was the fourth player. On entering the interview room, the experimenter introduced the interviewee to the interviewer and helped them both attach WiTilt v. 3.0 motion capture devices [35] to their wrists, head and torso using Velcro bands (see Measuring nonverbal coordination below). They were then sat facing one another either side of a low table.

Once set up, the experimenter gave the interviewer a set of questions about the first pre-interview task. A fixed question set was used to ensure consistency in the way interviewees' stories were examined (cf. [36,37]). The experimenter then retreated behind a screen to monitor the data capture. Once all questions about the first task were asked, the experimenter provided both the interviewee and the interviewer a post-task questionnaire. This questionnaire, which they completed independently, asked participants to indicate their agreement to a series of statements on a 'not at all' (1) to 'very much' (7) Likert scale. For the interviewee, five statements asked about the extent to which they agreed that: (i) the task was difficult; (ii) they were confident that they had performed well; (iii) they were anxious; (iv) the interaction was well-paced; and (v) the interaction was awkward. For the interviewer, six statements asked about the extent to which they agreed that: (i) the interviewee was telling the truth; (ii) the interaction was well-paced; (iii) the interaction was awkward; (iv) the interviewee was trustworthy; (v) the interviewee was honest; and (vi) the interviewee was suspicious. The first interviewer statement was used to determine lie detection accuracy. Once the ratings and veracity judgement were completed, this process was repeated for the second and third tasks.

The interview process described above was subject to two manipulations. First, half of the interviewees were asked to provide an account about playing a game of Cluedo in forward order, and half provided an account in reverse order. Previous research suggests that asking participants to recall an event in reverse order is more difficult, particularly for liars [32]. Thus, providing the account in forward order served as the 'difficult lie' condition, while providing the account in reverse order served as the 'very difficult lie' condition. Second, the order in which the topics were discussed at

interview was counterbalanced to test for order effects, either starting with the informal conversation or the Cluedo game. There were no significant order effects.

After interview, participants were debriefed and asked if they knew about the experiment's purpose. Although all participants knew from the use of the WiTilt devices that their nonverbal behaviour was being examined, none mentioned mimicry or coordination.

### 2.1.3. Measuring nonverbal coordination

We captured participants' body movement using WiTilt v. 3.0 motion capture devices [35]. A WiTilt is a matchbox-sized device that contains one 3-axis accelerometer, a gyroscope and a Bluetooth transmitter that enables wireless data transfer. The device is easily secured to a body part using Velcro without hindering natural movement. In this experiment, raw motion data from both interactants was obtained using four sensors each: one attached to the back of the head, one across the rib cage and one to each wrist. The WiTilts were set to broadcast raw sensor readings at 120 times a second (i.e. 120 Hz), which we collected using specially developed software (available at: https://github.com/sophievanderzee/DeceptionMimicry). Alongside the raw sensor output, the software enabled the experimenter to insert 'timestamps' into the data stream. These timestamps were used to identify the start and end of the questioning about each task.

Our analysis focuses on the gyroscope readings. These data measure the force enacted on the WiTilt device at any one time, and so they provide a concise single measure of the size, but not direction, of body part movement at each time point. For example, a large downward gesture would appear in the data as an increase then decrease in gyroscope values as the hand increases in motion then slows as it comes to the bottom of the gesture. These data represent a refinement in granularity of measurement often used in the manual coding of body movement (cf. [13]). These data were screened following the automated measurement and analysis of body motion method [38]. First, we removed the occasional data points whose value exceed the possible reading values for the sensor, because these represented electronic recording errors (approx. 0.5%). Second, because it was not possible to ensure that the devices began recording at the exact same moment, we aligned the data of each device by matching the timestamps associated with each data point to the first stamp that existed on both recordings. Third, to ensure that the streams remained aligned across the interaction, which may not have been the case once recording errors had been removed, we down-sampled the data to 5 frames $s^{-1}$. Fourth, because the absolute values of device outputs are influenced by their body placement, we produced standard scores of the data on each axis for each body part. For ease, these $z$-scores were multiplied by 1000, converting fractions into integers.

The screened data for interviewer–interviewee pairs were used to calculate a coordination score for each body part. Because coordinated movement occurs with a short time delay, and because that delay itself can vary in length across the interaction, we measured movement similarity using the novel approach of dynamic time warping (DTW). DTW may be thought of as a form of correlation that computes the optimal alignment of two sequences of measurements that may vary in time and speed. It is more flexible than cross-correlation and wavelet coherence methods of achieving this correlation because it allows the time scale of one sequence to be locally shortened or stretched in order to increase the alignment between measurement pairs. In other domains, this results in a more accurate identification of the similarity in shape between two time series [39,40]. The amount of compression and expansion undertaken by the analysis is limited by a 'step pattern' that describes the possible comparisons or 'steps' that may be taken for each data point. We used step pattern VI-c [41], which allows for moderate changes in the alignment in a way that is symmetrical (i.e. it does not make assumptions about which actor is 'leading' the coordination). Although our hypotheses are based on changes in interviewee's behaviour, we used a symmetrical measure in recognition of the mutually reinforcing nature of interaction, and to capture changes in nonverbal coordination as a whole. Given that the two sequences are aligned at the start and the end, a maximum offset between the two sequences is at one-sixth of the total length. However, in practice, the maximum offset would be in the order of a few seconds.

The final DTW score is then obtained as the average value of the pairwise distance between the two sequences at each aligned data point. A lower DTW score indicates greater similarity between the two data streams, reflecting the fact that participants are moving in more similar ways and thus less 'steps' are required to make the streams match. The lower the DTW score, the more similar movement of the dyad (i.e. more coordination), as measured at the granularity of 5 data points $s^{-1}$. To facilitate interpretation, we inverted the DTW scores (i.e. multiplied by –1.0) so that greater scores indicate more coordination. We then examined these scores across the interviewer–interviewee dyads for each

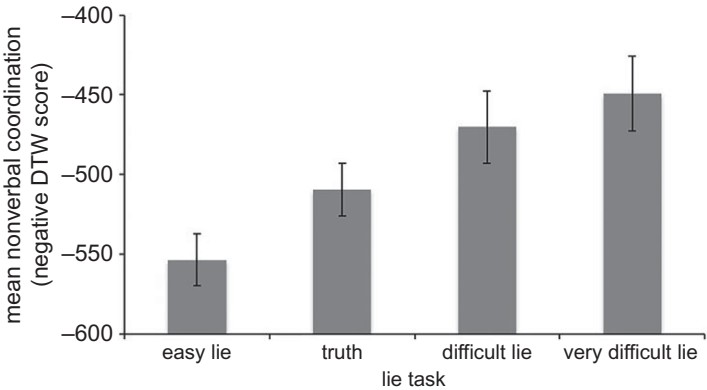

**Figure 1.** Mean nonverbal coordination (DTW) as a function of task. Error bars are 95% CI.

of the four body parts, and for an overall measure of coordination calculated as the average of the four body part DTW scores. Where a DTW score for one of the body parts was missing, we used the average of the available scores.

## 2.2. Results

### 2.2.1. Check of the task difficulty manipulation

The *post hoc* questionnaire responses provided the opportunity to check that the manipulation of lie type impacted interviewees' experiences. These data confirmed that our manipulation of task difficulty was successful, but for one exception: across all items, the interviewees reported the truth account as more difficult than telling the easy concealment lie. Specifically, regressions of task on, respectively, difficulty, confidence and anxiety revealed that interviewees reported: (i) increasing difficulty when telling the easy lie ($M = 2.37$, s.d. $= 1.50$), truth ($M = 2.79$, s.d. $= 1.57$), difficult lie ($M = 3.00$, s.d. $= 1.31$) and very difficult lie ($M = 4.05$, s.d. $= 2.01$), $F_{1,127} = 7.99$, $p = 0.005$, $r = 0.24$, 95% confidence interval (CI) (0.11, 0.65); (ii) decreasing confidence when telling the easy lie ($M = 5.67$, s.d. $= 1.44$), truth ($M = 4.79$, s.d. $= 1.73$), difficult lie ($M = 4.68$, s.d. $= 1.64$) and very difficult lie ($M = 3.76$, s.d. $= 2.14$), $F_{1,127} = 5.00$, $p = 0.027$, $r = 0.20$, 95% CI (0.04, 0.62) and (iii) increasing anxiety when telling the easy lie ($M = 2.51$, s.d. 1.35), truth ($M = 2.95$, s.d. $= 1.62$), difficult lie ($M = 3.05$, s.d. $= 1.33$) and very difficult lie ($M = 3.86$, s.d. $= 1.82$), $F_{1,127} = 4.50$, $p = 0.036$, $r = 0.19$, 95% CI (0.02, 0.53). These data suggest the appropriate ordering of task difficult should be easy lie, truth, difficult lie and very difficult lie. We use this order in the remainder of this paper.

A second method of checking the effects of task differences was provided by the interviewers' deception judgements, because previous research has shown that judges find it easier to identify liars under conditions of high compared to low cognitive load [32]. For each topic, interviewers were asked whether they thought the interviewee told the truth on a 7-point Likert scale ranging from 1 (not at all) to 7 (very much). A response was recorded as 'correct' when scoring 5–7 concerning truths and 1–3 concerning lies. A score of 4 (neutral) was considered incorrect. A $\chi^2$ analysis in which only lies were taken into account found that interviewers were significantly better at correctly identifying the very difficult lie (57% correct) compared to the difficult lie (32% correct) and easy lie (16% correct), $\chi^2_2 = 11.21$, $n = 86$, $p = 0.004$, $\Phi = -0.36$, 95% CI (0.16, 0.53). They were also more accurate at correctly identifying the informal conversation as truthful (63% correct) than they were at identifying lies overall (30% correct), $\chi^2_1 = 12.55$, $n = 129$, $p < 0.001$, $\Phi = 0.31$, 95% CI (0.15, 0.46), a finding that is consistent with a truth bias [42].

### 2.2.2. Interviewer–Interviewee nonverbal coordination

To test whether interviewer–interviewee coordination increases with greater cognitive load, we computed a mixed effect model in which task difficulty was a predictor of coordination and participant was entered as a random effect. The model, displayed in figure 1, revealed a linear increase in the amount of coordination with greater cognitive load, $F_{3,82} = 29.37$, $p < 0.001$, $r = 0.582$, 95% CI ($-0.686$, $-0.455$). Compared to their behaviour in the truthful task, interview dyads showed

greater coordination when interviewees told the difficult lie, $t_{21} = -2.71$, $p = 0.013$ (Bonferroni corrected $\alpha = 0.017$), $d = 0.578$, 95% CI ($-1.02$, 0.119), and the very difficult lie, $t_{19} = -4.38$, $p < 0.001$, $d = -0.980$, 95% CI ($-1.51$, 0.435), but the opposite occurred when interviewees told the easy lie, $t_{41} = 3.69$, $p = 0.001$, $d = 0.569$, 95% CI (0.240, 0.893). No coordination differences were found between the difficult and very difficult lie conditions, $F < 1$, non-significant.

To determine the generality of this finding, we repeated the analyses using the DTW scores per body part. The linear relationship between coordination and cognitive load was significant for all four body parts, torso, $F_{3,76} = 11.35$, $p < 0.001$, $r = -0.456$, 95% CI ($-0.589$, $-0.301$), head, $F_{3,82} = 20.96$, $p < 0.001$, $r = -0.435$, 95% CI ($-0.566$, $-0.282$), left hand, $F_{3,82} = 13.81$, $p < 0.001$, $r = -0.419$, 95% CI ($-0.552$, $-0.265$) and right hand, $F_{3,63} = 7.36$, $p < 0.001$, $r = -0.399$, 95% CI ($-0.553$, $-0.219$), indicating that this effect was generalized across multiple locations. To explore whether hand coordination was greater for ipsilateral (mirrored) movement than for isolateral movement, we also calculated equivalent DTW scores for interviewer left hand to interviewee right hand, and vice versa. The linear relationship between coordination and cognitive load was significant for interviewer left hand to interviewee right hand $F_{3,63} = 10.77$, $p < 0.001$, $r = -0.407$, 95% CI ($-0.560$, $-0.228$), and for interviewer right hand to interviewee left hand, $F_{3,63} = 9.53$, $p < 0.001$, $r = -0.425$, 95% CI ($-0.574$, $-0.248$).

To test whether interviewer–interviewee coordination increases with greater interviewer suspicion, we computed a mixed effect model in which task interviewer suspicion was a predictor of coordination and participant was entered as a random effect. The model showed that coordination did not increase owing to increased suspicion. $F_{1,84} = 0.523$, $p = 0.472$, $r = 0.064$, 95% CI ($-0.110$, 0.235).

## 2.3. Discussion

Our first experiment examined the relationship between nonverbal coordination and increasingly difficult deception tasks. Both upper-body and individual body part results suggest that interviewer–interviewee coordination increases with task difficulty, supporting the hypothesis (H1a) that the cognitive demands of telling difficult lies lead interactants to rely more heavily on nonverbal coordination. By contrast, the data are not consistent with the alternative possibility that the stress induced by difficult lies would lead to behavioural freezing, which would reduce the extent of coordination observed (H1b).

An alternative explanation for this finding is that participants monitored the interviewer more carefully when they were telling difficult and very difficult lies compared to when they were telling the truth and the easy lie. It is common for liars to report monitoring the responses of their interviewer to judge whether or not their account is believed [29,43]. Studies of facial mimicry [44] and behavioural mimicry [45] also imply that the association between attention and nonverbal coordination goes beyond the prerequisite of needing to see the other's behaviour to respond to it. For example, van Baaren et al. found that participants whose culture emphasized attention to the dyad showed greater nonverbal mimicry than those whose culture emphasized focus on self. Similarly, Santioni et al. [46] found that dyadic nonverbal coordination in 5 min conversations was predicted by protective self-monitoring, an individual difference that measures the extent to which a person is attentive and responsive to avoiding social loss within an interaction. Thus, a corollary of increased attention might be increased nonverbal coordination, such that the result of experiment 1 is a consequence not of cognitive load but of increased attention.

# 3. Experiment 2

To disentangle the influence of a liar's attention on nonverbal coordination, experiment 2 compared the behaviour of interviewees instructed to pay extra attention to the interviewer's nonverbal behaviour, interviewees instructed to pay extra attention to the interviewer's verbal behaviour, and a group that received no instruction. Using all three conditions allowed us to separate the effect of attention to nonverbal behaviour from the effect of increased cognitive load caused by adding the additional attention task, as well as the possibility that the attention instruction was interpreted generally and led participants to pay more attention to the interviewer's behaviour overall. If following an instruction increases cognitive load or attention in general, then coordination levels in the nonverbal and verbal instruction condition will be higher than coordination levels in the no instruction condition. By contrast, if attention to nonverbal behaviour is responsible for the coordination effect observed in experiment 1, then we should observe increased coordination in the nonverbal attention

condition but not in the verbal attention condition. Finally, if we observed no main effect or interaction efforts for the attention manipulation, but continue to find support for our prediction that behavioural coordination increases with task difficulty (H1a), then we can discard attention as an alternative explanation for the findings of experiment 1.

## 3.1. Method

### 3.1.1. Participants

Eighty-six students acted as interviewees or interviewers (age $M = 21.1$ years, range: 18–32). Interviewees ($n = 43$) received payment of £8 for 70 min of their time. Interviewers received £5 for 40 min of their time. One pair was excluded because of motion recording failure, leaving 42 pairs. Of these 42 pairs, the body movement data of three pairs, the head data of two pairs and the left- and right-hand data of one pair are missing due to sensor failure. A sample size of 15 participants per condition was established to be sufficient in a pilot study. The current experiment includes three between-subject conditions, leading to an aim of 45 pairs, of which we tested 43 pairs.

### 3.1.2. Procedure

Experiment 2 changed the design of experiment 1 in three ways. First, in order to manage the design complexity of the experiment, we removed the difficult lie condition. The three conditions are sufficient to enable comparisons across veracity and task difficulty. Second, we randomly allocated interviewees to one of three instruction conditions. These instructions, which were provided both verbally and in writing, asked interviewees to either: (i) 'pay extra attention to the nonverbal behaviour of the interviewer ($n = 12$)'; (ii) 'pay extra attention to the verbal behaviour of the interviewer ($n = 14$),' or (iii) they did not receive an instruction ($n = 16$). Third, in order to check the attention manipulation, we used an extended version of the post-task questionnaires. For interviewees, the additional items asked about 'the extent to which they experienced difficulty following the instructions (1 = 'not at all' to 7 = 'very much so'),' and 'the amount of attention they paid to 'the interviewer's nonverbal behaviour' (1) versus 'the interviewer's verbal behaviour' (7).' For interviewers, the additional items asked 'how much they felt the interviewee was distracted,' and 'how much attention they felt the interviewee was paying to their nonverbal and verbal behaviour.'

### 3.1.3. Measuring nonverbal coordination

The recording and analysis of nonverbal coordination was identical to experiment 1.

## 3.2. Results

### 3.2.1. Check of the task difficulty manipulation

As in experiment 1, our *post hoc* questionnaire enabled us to confirm that the task manipulation systematically affected interviewees' experiences of difficulty. Interviewees reported: (i) increasing difficultly when telling the truth ($M = 2.00$, s.d. = 1.21), easy lie ($M = 2.62$, s.d. = 1.40) and very difficult lie ($M = 4.45$, s.d. = 1.61), $F_{2,82} = 45.90$, $p < 0.001$, $\eta_p^2 = 0.528$, 95% CI (0.369, 0.627); (ii) decreasing confidence when telling the truth ($M = 5.57$, s.d. = 1.29), easy lie, ($M = 5.40$, s.d. = 1.27) and very difficult lie, ($M = 3.57$, s.d. = 1.48), $F_{2,82} = 33.69$, $p < 0.001$, $\eta_p^2 = 0.451$, 95% CI (0.283, 0.563); and (iii) increasing anxiety when telling the truth ($M = 2.38$, s.d. = 1.38), easy lie ($M = 2.60$, s.d. = 1.31) and very difficult lie ($M = 4.31$, s.d. = 1.62), $F_{2,82} = 35.58$, $p < 0.001$, $\eta_p^2 = 0.465$, 95% CI (0.297, 0.574).

As before, interviewers were asked to judge the veracity of interviewees' statements about each task. In line with the detection rates from experiment 1, a significant association was found between lie task and detection rate, $\chi_1^2 = 4.94$, $n = 84$, $p = 0.026$, $\Phi = 0.24$, 95% CI (0.03, 0.44). Interviewers more often correctly identified the very difficult lie (52% correct) compared to the easy lie (29% correct), and overall, they were more accurate at correctly identifying truths (74% correct) than they were at identifying the lies in general (40% correct), $\chi_1^2 = 12.46$, $n = 126$, $p < 0.001$, $\Phi = 0.31$, 95% CI (0.15, 0.47).

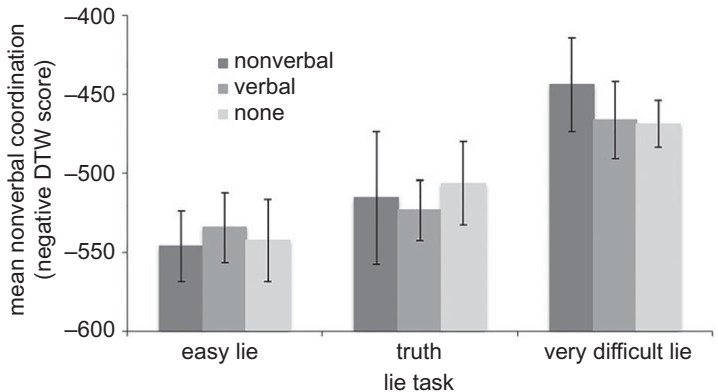

**Figure 2.** Average nonverbal coordination (DTW) as a function of task and attention instruction. Error bars represent 95% CI.

### 3.2.2. Check of the attention manipulation

The post-task questionnaires confirmed that the instruction led to differences in the experiences of the interviewee and interviewer. A 3 (instruction) × 3 (task) mixed ANOVA revealed that self-reported attention for the interviewer was influenced by attention instruction, $F_{2,39} = 4.33$, $p < 0.020$, $\eta_p^2 = 0.182$, but not by task, $F_{2,78} = 0.08$, $p = 0.926$, $\eta_p^2 = 0.002$. Interviewees who received the nonverbal instructions reported paying more attention to the interviewer's nonverbal behaviour ($M = 3.53$, s.d. = 1.30) compared to interviewees in the verbal condition ($M = 4.55$, s.d. = 1.21); $t_{76} = -3.58$, $p = 0.001$ (Bonferroni corrected $\alpha = 0.025$), $d = 0.81$, 95% CI (0.34, 1.28), and no instruction condition ($M = 4.67$, s.d. = 1.37), $t_{82} = -3.85$, $p < 0.001$, $d = 0.85$, 95% CI (0.39, 1.31). A Pearson correlation showed that both instruction condition (i.e. the amount of attention interviewees were supposed to pay), $r = -0.02$, $n = 124$, $p = 0.870$, and how much self-reported attention the interviewee had for the interviewer's verbal and nonverbal behaviour did not affect the observed nonverbal coordination, $r = 0.13$, $n = 126$, $p = 0.172$.

Interviewers' ratings of their perceptions of the interviewees' attention provided the opportunity to check whether interviewers were aware of the instruction given to the interviewees. Two 3 (instruction) × 3 (task) mixed ANOVAs were performed with the interviewer's perceptions of the interviewee's attention for their nonverbal and verbal behaviour. For both, task type and instruction did not affect the interviewer's perceptions, indicating that interviewers were not aware of the attention instructions given to the interviewees, all $F < 1$, non-significant.

### 3.2.3. Interviewer–Interviewee nonverbal coordination

The new difficulty of task order of easy lie, truth and very difficult lie was used to measure the influence of task and attention instructions on interviewer–interviewee coordination. Figure 2 shows upper-body coordination as a function of instruction and task. A 3 (attention instruction) × 3 (task) mixed ANOVA with task as the repeated measure and DTW score as the dependent variable revealed a main effect for task, $F_{2,76} = 48.14$, $p < 0.001$, $\eta_p^2 = 0.559$, $r = 0.748$, but not for attention instruction, $F_{2,38} = 0.09$, $p = 0.914$, $\eta_p^2 = 0.005$, $r = 0.070$, and no interaction between task and attention instruction, $F_{4,76} = 0.32$, $p = 0.316$, $\eta_p^2 = 0.060$, $r = 0.245$. Regardless of attention instruction, more coordination occurred during the very difficult lie compared to the easy lie, $t_{40} = 9.29$, $p < 0.001$ (Bonferroni corrected $\alpha = 0.017$), $d = 1.45$, 95% CI (1.01, 1.89), and compared to the truth, $t_{40} = 6.64$, $p < 0.001$, $d = 1.04$, 95% CI (0.65, 1.41). In addition, interviewees mimicked more when telling a truth compared to telling an easy lie, $t_{40} = 3.01$, $p = 0.004$, $d = 0.471$, 95% CI (0.15, 0.79).

To examine the cause of task difficulty on nonverbal coordination in more detail, four equivalent ANOVAs were conducted for head, torso and wrist movement. They revealed an equivalent body part-specific increase in coordination, with a significant task effect for movement of all body parts: torso, $F_{2,72} = 11.05$, $p < 0.001$, $\eta_p^2 = 0.235$, $r = 0.485$; head, $F_{2,74} = 22.27$, $p < 0.001$, $\eta_p^2 = 0.376$, $r = 0.613$; left hand, $F_{2,74} = 22.05$, $p < 0.001$, $\eta_p^2 = 0.373$ $r = 0.611$; and right hand, $F_{2,74} = 26.90$, $p < 0.001$, $\eta_p^2 = 0.421$, $r = 0.649$.

## 3.3. Discussion

Experiment 2 manipulated the extent to which participants paid attention to verbal and nonverbal behaviour to determine whether changes in attention accounted for the increase in nonverbal

coordination observed in experiment 1. Although interviewees reported paying more attention to the behaviour identified in their instructions, this did not affect the degree of interviewer–interviewee coordination. Instead, nonverbal coordination increased with task difficulty irrespective of attentional focus. This rules out the alternative explanation that coordination increased owing to interviewees' raised attention towards their interviewer when delivering difficult lies. Moreover, both interviewees' self-reported attention levels and interviewers' awareness of attention paid to them by interviewees were not influenced by task (i.e. when telling truths, easy and very difficult lies). This suggests that, in contrast to the suggestion of interpersonal deception theory [43], liars do not monitor their interaction partner more closely than truth-tellers, or at least they are not consciously aware of doing so. However, our evidence is not definitive on this issue. For example, the task of presenting two lies and a truth may have prompted participants to adopt a 'deceptive' approach to monitoring across all of their accounts. It may be the case that participants' reporting does not reflect what they did during the interaction. These possibilities will need addressing in future work, for example, by tracking eye movements of people telling truths and lies.

# 4. General discussion

Taking advantage of motion capture technology, the two studies presented in this paper are among the first to examine interpersonal aspects of deceptive behaviour. Their results provide evidence that nonverbal coordination increases with lie difficulty, and that neither interviewer suspicion nor increased attention towards nonverbal behaviour causes this effect. This is consistent with the proposition that mimicry increases under cognitive load [20] because of the increased reliance on the automatic processes of interpersonal behaviour. This finding is consistent with the broader notion that automated processes may become more prevalent when cognitively loaded, which has previously been demonstrated in the context of decision-making. For example, the dual-process approach to decision-making proposes that people tend to rely more heavily on the use of heuristics and other intuitive 'system 1' processing when depleted, compared to relying on the more deliberate and effortful 'system 2' processes when not depleted [47,48].

To our knowledge, this is the first research to show that cognitive load impacts nonverbal coordination in an applied context that complements previous tests with isolated body parts such as finger movement [20]. The possibility that automatic mimicry [14] is the root cause of the behaviour we observed makes it an interesting, and thus far unexplored, cue to deceit, especially given that people's attempts to control their lying behaviour is one of the main obstacles in the search for universal nonverbal cues to deceit [16,21]. When attempting to appear credible, liars may implement countermeasures such as avoiding behaviours related to lying, or deliberately showing behaviours associated with honesty [29,49]. Arguably, it would be harder to successfully manipulate automatic behaviours in order to appear credible, but to what extent liars can effectively deploy countermeasures against the use of nonverbal coordination as a cue to deceit—and indeed whether a user could distinguish deceit from statement difficulty using the cue—will need to be tested.

By observing the natural occurrence of behaviour during interaction, we were able to examine changes in nonverbal coordination without the interactants' awareness. Although this approach probably captured automatic aspects of interpersonal behaviour, our findings cannot exclude the occurrence of a more conscious form of coordination. For example, Guéguen [50] demonstrated that the deliberate mimicking of another person's nonverbal behaviour can reduce the need for deception detection, because it elicits more honest behaviour in the mimicked person. Dunbar *et al*. [51] proposed that when lying voluntarily, interviewees use coordination deliberately as a social strategy to appear credible. The consequence of these efforts may be reciprocal coordination by the interviewer, which has the effect of decreasing an interviewer's ability to identify deceit [13]. However, the lack of association between coordination and interviewer suspicion in our experiment provides evidence that liars were not consciously increasing their mimicking behaviour when met with suspicion. Moreover, when asked directly at debrief, none of our participants, including interviewers, mentioned deliberately using mimicry, stressing the unconscious, automatic aspects of nonverbal coordination in this setting. Future research can establish whether instructing interviewers to focus on nonverbal coordination or mimicry occurrence will increase detection rates, or whether these effects are too subtle to be captured without the use of technology.

Interestingly, while interviewees were consistent in how much they matched behaviour during truth telling and concealing information, their perception of task difficulty was not consistent across the two studies. While in the first experiment interviewees reported finding being truthful more difficult than

concealing their cheating (i.e. the easy lie), in the second experiment, this effect was found in the opposite direction. This suggests that lying might not always be more difficult than truth telling, particularly when the lie involves a simple concealment rather than a fabrication [30]. Our findings are consistent with the possibility that in some social situations concealing information or telling a white lie can be socially beneficial and less hard than telling the truth [16,52]. Because cheating on a task is seen as dishonest behaviour, and people tend to negatively judge dishonesty, concealing this transgression may indeed be easier than being truthful [53]. This possibility highlights the need for combining a careful design of the deception condition within studies with robust measurement of the difficulty experienced by liars and truth-tellers. The absence of effects seen in prior studies could conceivably be owing to a false assumption regarding the relative difficulty of the lie and truth condition for the participants.

Traditionally, nonverbal coordination and mimicry is measured by manually annotating isolated movements from selected body parts based on video data (cf. [13]). The use of motion capture devices in this paper provides an efficient measure of nonverbal coordination that is less susceptible to the reliability issues associated with manual coding [38,54] and the limitations of measuring time course associated with social experimentation [19]. Our operationalization through DTW scores provides an overall score of the amount of coordination between both interactants. In this paper, we have not analysed leading and following patterns, which would require a parametric approach to take into account, and to compensate for, issues such as maximum delays and recurrence and absence of movement. Thus, we can only determine whether the degree of coordination has changed, and not whether the interviewer or interviewee was leading this change. Identifying who is leading the increase in coordination occurrence is an interesting avenue for future research.

## 5. Conclusion

In two experiments, we found that the coordination of interviewer–interviewee nonverbal behaviour increased as the interviewee's task became more cognitively demanding. This finding is consistent with the theory that task difficulty reduces the cognitive resources an interviewee has available to manage their social behaviour. Because our experiments used WiTilt devices focused on just four body parts, we cannot conclude for certain that the interviewees engaged in automatic imitation of the interviewer's behaviour. Researchers keen to build on our initial evidence and examine this question might usefully use a full-body motion capture system [55], or even accurate remote systems such as Microsoft Kinect and automated video analysis that will further increase the ecological validity and the applicability of automatic analysis [56]. Such advances in motion capture technologies are allowing researchers to test the role of mimicry in interpersonal cooperation and deception in increasingly unconstrained paradigms [57,58]. The result is that we are beginning to understand how basic interpersonal processes play out in complex, real world scenarios, such as the interview room.

Ethics. Experiments were approved by Lancaster University's Research Ethics Committee, and is in line with the World Medical Association Declaration of Helsinki. All participants gave informed consent in writing before taking part.

Data accessibility. The data, methods and software are openly available at https://github.com/sophievanderzee/DeceptionMimicry.

Authors' contributions. S.V.D.Z., P.J.T., J.D. and R.W. developed the study concepts and design. T.M. created the software needed to capture the data. R.W. and S.V.D.Z. collected the data. S.V.D.Z. and P.J.T. performed data analysis and drafted the manuscript together. All authors provided feedback and approved the final version of the manuscript for submission.

Competing interests. We declare we have no competing interests.

Funding. Centre for the Protection of National Infrastructure grant awarded to P.T. and PhD studentship awarded to S.V.D.Z., and European Research Council Starting grant 638408 Bayesian Markets.

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
