## [Reviewer comments · Royal Society Open Science]

Review History

RSOS-200839.R0 (Original submission)

Review form: Reviewer 1 (Gordon Wright)

Is the manuscript scientifically sound in its present form?

Yes

Are the interpretations and conclusions justified by the results?

Yes

Is the language acceptable?

Yes

Do you have any ethical concerns with this paper?

No

Have you any concerns about statistical analyses in this paper?

No

Recommendation?

Accept as is

Comments to the Author(s)

RSOS-200839 Reviewer Comments - Hope I'm not Reviewer 2!

Is the manuscript scientifically sound in its present form? Yes

Are the interpretations and conclusions justified by the results? Yes

Is the language acceptable? Yes

Is it clear how to access all supporting data? Yes

Do you have any ethical concerns with this paper? No

Have you any concerns about statistical analyses in this paper? No

The current paper presents 2 experiments, each examining the association between deception and the nonverbal mimicry of interviewer and interviewee. Manipulations are employed which increase cognitive load involved in the deception attempts, and, in the second experiment, attention is drawn to verbal and nonverbal behavioural channels on the part of the would-be liar. The results suggest that as cognitive load increases, the 'automatic' mimicry inherent in interpersonal communication between interviewer and interviewee increases.

I find the two experiments to be delivered with care and attention to detail, employing a paradigm which appears well-considered. The researchers have clearly attended to ethical concerns in the most diligent fashion and the presentation of data, results and analysis is top notch. I am also pleased to commend the researchers for avoiding any unnecessary complexity with regards to the body motion data and interpretation.

I thank the researchers for supplying sufficient information to repeat the analysis and for sharing their raw perception data and body motion data via github.

Page 3, Line 49 – I find the opening paragraphs well-written and compelling. They summarise the background literature clearly. It has been a while since I read Shaw et al., 2015 and query whether the claims of increased information elicitation and deceit cue elicitation are entirely justified based on the results of that paper. My recall is that the results are not quite that clear-cut. And in that paper, if memory serves, it is interviewers who mimic the interviewees.

Experiment 1

Page 6, Line 10 - Respectable sample size on the face of it. I'd be interested to see more of the information on sample size estimate (such as effect size achieved in the Pilot study mentioned – page 6, line 29).

Page 6, Line 45 – Initially unclear why the design is mixed with truth/concealment (WS) and the difficult / very difficult lie (BS). A quick comment on this would help. Even by page 9, line 29 this is somewhat unclear as to why this would be designed as such. Quick comment.

Page 7, Line 31 – Confederate task query

Page 7, Line 54 – I like the design of the confederate task in the difficult lie condition. It achieves the blend of a clear task, with achievable outcome, but with sufficient variability/detail and clearly an aspect of motivation by incentive. I often work with students and collaborators who are keen to develop deception tasks of this type, with a view to operationalising deception with the goal of eliciting certain types of behaviour. If space permits, I'd love to see a couple of points from the authors on the key features of the task that illustrate the thought put into the development of the task. I think it would be useful for other researchers interested in this type of experimental development.

Page 11, Line 22 – It occurs to me at this point that no mention of 'direction' of mimicry has been suggested up front explicitly, although since the would-be liar is the participant under increased cognitive load, and under such conditions mimicry appears to increase, it might be fair to assume the naïve assumption that the liar mimics the interviewer, or am I wrong? The symmetry decision within DTW therefore seems to require some support – or is this a more liberal approach than assuming Interviewer behavioural cues preceding the interviewees mimicry.... Or maybe the specific nature/direction of the mimicry (or any debate around it) is worth highlighting in the

introduction perhaps? I'm not fully up to speed with the mimicry literature, so it may be that distinctions around sequence or 'direction' are ignorant, but I think a naïve reader would welcome information on this. [I do see that you deal with some of these points in the General Discussion - nicely]. I still wonder if this could be pre-empted somewhat.

Page 11, line 31 - I am aware of the expertise of the research group with regards to the analysis of NVB, so please bear with me - these may be silly questions. I am keen to understand what the DTW scores will actually be telling me and at what level of granularity. Synchrony? Magnitude? Is there any opportunity to explore the patterns in the various limbs. I'm interested to know if there is spatial matching etc.

Page 12, Line 3 - I am not surprised by the finding that the truth-telling might be more difficult than the concealment lie (and closely aligned confidence & anxiety ratings). It's commonly stated that lying is more difficult than telling the truth, which is clearly not the case here, so perhaps a short riff on this in the discussion is merited? [Stet. I see that you did touch on this in the discussion]. I certainly worry that similar patterns of results may be sitting in file drawers as a simple comparison of Lie v Truth would appear null. Good.

Page 12, Line 31 - Deception judgements were recorded using likert scale responses to "to what degree the interviewee was telling the truth" page 9, line 8 and then you present X2 analyses. So you categorised responses on the 1-7 scale? A quick comment on how that categorisation was achieved? Just for my sanity.

Experiment 1 results:

Generally clear presentation. Thank you.

Page 14, line 3 - I appreciate the unpacking of each limb results at page 14, line 3. As I think I asked above, I was curious if the averaging across limbs might have been problematic, in that perhaps it illustrates gross postural shifts or something other than motor mimicry. Am I right in thinking that this suggests limb to limb specificity? And if so, did you mention the orientation of interviewer/interviewee? And is there any pattern between the hands? Ipsilateral (if memory serves on the terminology)...

Page 14, line 31 - As this is a within-subject manipulation of veracity, I find it unclear when you speak of liars and truth tellers, and would prefer discussion of condition specific differences. I'm having trouble now reconciling the results with the concealment lie and lowest level of mimicry, then truth showing increased mimicry. Does this fit with the alternative explanation? Maybe it does. A quick comment on this might help me.

Experiment 2

Page 15, line 3 - I like the attention manipulations and use of two control groups to monitor the effect of added attentional demands to an alternative channel. But again, this suggests that interviewees will be mimicking the interviewer, which entirely makes sense, but I feel like I had to do quite a bit of work to be confident in that not being silly. My suggestion would be to cover off this potential misunderstanding in the introduction.

Page 18, line 29. So the results persist, whereby more mimicry is seen under conditions of increasing cognitive load. Nice.

Page 19, line 44 - I tend to agree with the assertion that the 2nd experiment covers off the alternative explanation. Nicely played.

Page 19, line 53 - I was wondering if you'd touch on the IDT assertion that when lying, individuals might attend more to the target to monitor suspicion etc. I think that the current task isn't designed to fully test this assertion, and I'd worry that an individual, who is required to deliver at least 2 lies and a truth in random order, might adopt a 'deceptive' approach throughout - which would give the same pattern of results you present, no? I don't have a problem with the suggestion you present, but it slightly jumped off the page as unnecessary.

General Discussion

Page 20, line 29 - Is there a typo? "in the context OF decision-making" ??

Page 20, line 31 – I'd probably pop a pair of " around 'System 1' and 'System 2'. But I'm not fully on top of the RS style guide.

Page 20, line 47. Attempts plural? Something catches in the first part of the line.

Page 21, line 6 – Is it 'to what extent liars can'...? And again, I urge care in the suggestion that monitoring of motor mimicry might lend itself to lie detection efforts without carefully highlighting that even the results presented here don't immediately discriminate honesty from deception in any clear way. The easy lie obviously spoils that party. Not to say that I disagree though!

Page 22, line 17 – I'm pleased to see the discussion of easy truth/difficult lie assumption – and data to confound this assumption! I was thinking that a concealment in this scenario would also be easier by dint of the fact that it is a lesser communicative task; there is no fabrication involved (as per design) and likely just a denial of 'anything odd having happened'... I do however believe that telling a lie about nothing odd having happened (if it had) would be more difficult than telling the truth about nothing odd having happened (if it hadn't)... Obviously, this is me just speculating.

Final comments. Very thoughtful closure. Enjoyed this paper very much.

Review form: Reviewer 2

Is the manuscript scientifically sound in its present form?

No

Are the interpretations and conclusions justified by the results?

No

Is the language acceptable?

Yes

Do you have any ethical concerns with this paper?

No

Have you any concerns about statistical analyses in this paper?

No

Recommendation?

Major revision is needed (please make suggestions in comments)

Comments to the Author(s)

GENERAL

A general comment refers to the concept of mimicry. Could the authors please define it clearly and define their operationalization clearly – early on. This is crucial because mimicry is a form of interpersonal coordination, like social synchrony and the both are sometimes used interchangeably. But mimicry specifically implies imitation and copying. Therefore, an evaluation of whether the DTW is an appropriate method to quantify mimicry and whether an interpretation of findings in relation to "automatic imitation" is warranted, depends on the author's definition. A general switch to a more general "coordination" terminology instead of "mimicry" would be in acceptable, otherwise, I would like to ask the authors to offer a definition and operationalization of mimicry in their study and justifications for the use of the DTW and the interpretation of results that would be consistent with their definition of mimicry.

Another more general comment refers to the need for a clearer and more detailed explanation of the use of gyroscope data with the DTW method.

Finally, I find that Experiment 2 has a design that does not manage 100% to manipulate attention and cognitive load in the way it was intended and I would like more clarification from the authors or a discussion of this as a limitation.

More comments and details follow below:

ABSTRACT

1. The abstract mentions Experiment 2 but Experiment 1 should be clearly named as well. DTW as a method should also be mentioned in the abstract

INTRODUCTION

2. I recommend a clear introduction of terminology of terms like "interviewer" and "interviewee = liar"

3. The link between lying as a stressor and freezing is not convincing and a reference for "and may subsequently lead to freezing" could help.

4. I suggest that the introduction finishes with a clear summary of the research gaps and therefore a phrasing of the research aim(s). The second to last paragraph in the introduction does this well and leads the reader to understand why cognitive load matters. But the final paragraph ends too abruptly.

5. The authors mention in the second to last paragraph that there are two opposing ways in which lying may affect mimicry. But it is unclear what opposing means. I have attempted a summary and if this is correct I suggest that more signposting is used to make this clearer in the text: Way 1: Lying is cognitively demanding hence will lead to increased mimicry. Way 2: Lying is stressful which can lead to freezing hence decreased mimicry. A good way to signpost would be to actually state the theory first before then deepening the rationale.

6. Because the second to last paragraph is so long, I recommend starting the last one with more signposting like "The other way in which lying may affect mimicry is related to..."

7. This sentence and reference are unclear "In humans, mimicry can facilitate information elicitation and elicit cues to deceit (Shaw et al., 2015), could the. Authors explain what they mean?"

8. The introduction should have a clear literature discussion on the link between mimicry and attention. The discussion of Exp 1 says "This positive association between mimicry and attention has previously been observed in relation to facial mimicry (Likowski, Muhlberger, Seibt, Pauli, & Weyers, 2008)." But this would need to be elaborated and ideally in the introduction, because that sets up both studies in the main body

EXPERIMENT 1

9. Please explain very briefly the terminology of fabrication and lying early on as the terms are introduced, for example also reverse fabrication. This becomes evident further on, but may be confusing the first time they are mentioned to readers who are not versed in the deception literature

10. Please formulate clear hypotheses that can then be referred to in the discussion

11. "A minimum sample size of 15 participants per condition was established in a pilot study." How was this established? Can the authors give more details regarding the power analysis?

12. What is the difference between the difficult and very difficult tasks? The reverse order becomes clear later in the text but should be clarified as early as possible when the authors describe the Pre-interview Tasks, under "Difficult and very difficult lie task"

13. What if participants were already familiar with the game Clue compared to completely naïve participants. Do the authors think this might be a source of variability, could this be controlled for?

14. Please explain what was the cover story used to explain to participants why they are being made to wear the sensors?

15. The authors briefly mention that because it was not possible to ensure that the devices began recording at the exact same moment, they aligned the data of each device. Please explain how this alignment was achieved.

16. How were missing data handled? For example, if head movement is missing in a pair but all other sensors aren't? I assume that when reporting the individual body parts, the sample size varies, but what about the average score?

17. Does DTW take into account only timing, or angle size as well? So, if the authors are looking at gyroscope data, what exactly are they using as raw data for the DTW analysis?

18. Can the authors give evidence of DTW being used in either mimicry or joint action literature and if yes with what kind of data? Please make it clear if this is something previously validated or something exploratory and new. In the latter case it would be good to have a justification for including this method that shows how this is advantageous by comparison to previous (e.g. cross-correlation, wavelet coherence, cross-recurrence) methods.

19. We averaged the four DTW scores per limb - unclear what are the four scores? Do you mean body part instead of limb? You had head, rib cage and 2 wrists. Correct this for the rest of results/discussion section. Limbs mean arms or legs only

RESULTS

20. Is there a correction for multiple comparisons applied when performing the pairwise comparisons?

DISCUSSION

21. Please see my general comment on mimicry vs coordination/synchrony - it matters what we call it. Because mimicry is imitation, whereas a general synchrony or coordination does not necessarily entail imitation at all. And this will affect how the authors interpret the results

22. Note that the way the introduction is set up, it leads the reader to expect a discussion of the idea around how stressful the lying is perceived/freezing. This should be discussed here and how this Exp 1 may be used to corroborate one or the other opposing theories.

EXPERIMENT 2

23. If liars are thought to monitor the partners more than truth tellers, this means they pay more attention to them. But is there any reason to think that this would be the case for nonverbal monitoring and not also general monitoring, including verbal monitoring? This is important because it influences the authors' hypotheses for Experiment 2 and it is not clear why they hypothesize "If following instructions increases cognitive load in general, then mimicry levels in the nonverbal and verbal instruction condition will be higher than mimicry levels in the no instruction condition"). And then we have the factor related to attention ("nonvb", "verbal" and "none"), and we'd expect a replication of findings in the "none" condition. But then, also, depending on the hypothesis, some interactions. I find it hard to map this design and hypotheses onto what has been actually done and I would like more clarifications.

24. "First, in order to manage the design complexity of the experiment, we removed the difficult lie condition." I find that if Experiment 2 was done as a follow-up experiment, two task loads would have been enough, e.g. "easy lie and difficult fabrication" to make sure the choice is of those tasks that showed clearest difference. Why include or exclude anything else?

25. Although explained in different ways, it is unclear how many sample pairs there were per condition? Please clarify.

26. I find it difficult to believe that the instruction "pay extra attention to the nonverbal behaviour of the interviewer" is enough to create attention AND cognitive load. Also, the check in the questionnaire, is too vague "the amount of attention they paid to 'the interviewer's nonverbal behaviour' (1) versus 'the interviewer's verbal behaviour' (7)." Why not ask them to

count something specific in their partner's nonverbal behavior like the raising of the left arm, or ask them to estimate the speed of the movement. Otherwise, the claim "Using both control groups allowed us to separate the effect of attention to nonverbal behaviour from the effect of increased cognitive load caused by adding the additional task of having to follow the attention instructions." is not 100% convincing

27. "This suggests, in contrast to previous assertions (Buller & Burgoon, 1996; Kassin & Gudjonsson, 2004; Schweitzer et al., 2002), that interviewees do not increasingly monitor their interaction partner when lying, or at least they are not consciously aware of doing so." How did the measure in this study differ to those referenced studies? This study included very explicit questions to measure a very implicit phenomenon, I wonder if an implicit phenomenon may not be best captured by an implicit measure? Perhaps eye-tracking might be a good implicit measure to be used? Again, what did the authors in the referenced papers use?

GENERAL DISCUSSION

28. Please add a conclusion

OTHER

29. The supplementary material contains some redundant information with regard to the DTW (i.e. it is explained in the main manuscript text).

Decision letter (RSOS-200839.R0)

Dear Dr Taylor,

The editors assigned to your paper ("A Liar and a Copycat: Nonverbal Mimicry Increases with Lie Difficulty") have now received comments from reviewers. We would like you to revise your paper in accordance with the referee and Associate Editor suggestions which can be found below (not including confidential reports to the Editor). Please note this decision does not guarantee eventual acceptance.

Please submit a copy of your revised paper before 12-Aug-2020. Please note that the revision deadline will expire at 00.00am on this date. If we do not hear from you within this time then it will be assumed that the paper has been withdrawn. In exceptional circumstances, extensions may be possible if agreed with the Editorial Office in advance. We do not allow multiple rounds of revision so we urge you to make every effort to fully address all of the comments at this stage. If deemed necessary by the Editors, your manuscript will be sent back to one or more of the original reviewers for assessment. If the original reviewers are not available, we may invite new reviewers.

When submitting your revised manuscript, you must respond to the comments made by the referees and upload a file "Response to Referees" in "Section 6 - File Upload". Please use this to

document how you have responded to the comments, and the adjustments you have made. In order to expedite the processing of the revised manuscript, please be as specific as possible in your response.

- Data accessibility

If you wish to submit your supporting data or code to Dryad (<http://datadryad.org/>), or modify your current submission to dryad, please use the following link:
<http://datadryad.org/submit?journalID=RSOS&manu=RSOS-200839>

- Competing interests

- Authors' contributions

- Acknowledgements

- Funding statement

on behalf of Dr Carina de Klerk (Associate Editor) and Essi Viding (Subject Editor)
openscience@royalsociety.org

Associate Editor's comments (Dr Carina de Klerk):

Comments to the Author:

Dear authors,

The reviewers have provided extensive comments and raised a number of important points. In particular, clarification is needed regarding whether the DTW analyses measured mimicry or general coordination/behavioural synchrony, more information needs to be provided about the sample size per group and power analyses, and about the DTW analyses. Additionally, reviewer 2 raises valid concerns about the manipulation and measurement of attentional monitoring in Experiment 2.

I believe that the reviewers' comments should be addressable and I therefore invite you to respond them and revise your manuscript, which will be sent back to review.

Sincerely,
Carina de Klerk

Comments to Author:

Reviewers' Comments to Author:

Reviewer: 1

Comments to the Author(s)

RSOS-200839 Reviewer Comments - Hope I'm not Reviewer 2!

Is the manuscript scientifically sound in its present form? Yes

Are the interpretations and conclusions justified by the results? Yes

Is the language acceptable? Yes

Is it clear how to access all supporting data? Yes

Do you have any ethical concerns with this paper? No

Have you any concerns about statistical analyses in this paper? No

The current paper presents 2 experiments, each examining the association between deception and the nonverbal mimicry of interviewer and interviewee. Manipulations are employed which increase cognitive load involved in the deception attempts, and, in the second experiment,

attention is drawn to verbal and nonverbal behavioural channels on the part of the would-be liar. The results suggest that as cognitive load increases, the 'automatic' mimicry inherent in interpersonal communication between interviewer and interviewee increases.

I find the two experiments to be delivered with care and attention to detail, employing a paradigm which appears well-considered. The researchers have clearly attended to ethical concerns in the most diligent fashion and the presentation of data, results and analysis is top notch. I am also pleased to commend the researchers for avoiding any unnecessary complexity with regards to the body motion data and interpretation.

I thank the researchers for supplying sufficient information to repeat the analysis and for sharing their raw perception data and body motion data via github.

Page 3, Line 49 - I find the opening paragraphs well-written and compelling. They summarise the background literature clearly. It has been a while since I read Shaw et al., 2015 and query whether the claims of increased information elicitation and deceit cue elicitation are entirely justified based on the results of that paper. My recall is that the results are not quite that clear-cut. And in that paper, if memory serves, it is interviewers who mimic the interviewees.

Experiment 1

Page 6, Line 10 - Respectable sample size on the face of it. I'd be interested to see more of the information on sample size estimate (such as effect size achieved in the Pilot study mentioned - page 6, line 29).

Page 6, Line 45 - Initially unclear why the design is mixed with truth/concealment (WS) and the difficult / very difficult lie (BS). A quick comment on this would help. Even by page 9, line 29 this is somewhat unclear as to why this would be designed as such. Quick comment.

Page 7, Line 31 - Confederate task query

Page 7, Line 54 - I like the design of the confederate task in the difficult lie condition. It achieves the blend of a clear task, with achievable outcome, but with sufficient variability/detail and clearly an aspect of motivation by incentive. I often work with students and collaborators who are keen to develop deception tasks of this type, with a view to operationalising deception with the goal of eliciting certain types of behaviour. If space permits, I'd love to see a couple of points from the authors on the key features of the task that illustrate the thought put into the development of the task. I think it would be useful for other researchers interested in this type of experimental development.

Page 11, Line 22 - It occurs to me at this point that no mention of 'direction' of mimicry has been suggested up front explicitly, although since the would-be liar is the participant under increased cognitive load, and under such conditions mimicry appears to increase, it might be fair to assume the naïve assumption that the liar mimics the interviewer, or am I wrong? The symmetry decision within DTW therefore seems to require some support - or is this a more liberal approach than assuming Interviewer behavioural cues preceding the interviewees mimicry.... Or maybe the specific nature/direction of the mimicry (or any debate around it) is worth highlighting in the introduction perhaps? I'm not fully up to speed with the mimicry literature, so it may be that distinctions around sequence or 'direction' are ignorant, but I think a naïve reader would welcome information on this. [I do see that you deal with some of these points in the General Discussion - nicely]. I still wonder if this could be pre-empted somewhat.

Page 11, line 31 - I am aware of the expertise of the research group with regards to the analysis of NVB, so please bear with me - these may be silly questions. I am keen to understand what the DTW scores will actually be telling me and at what level of granularity. Synchrony? Magnitude? Is there any opportunity to explore the patterns in the various limbs. I'm interested to know if there is spatial matching etc.

Page 12, Line 3 - I am not surprised by the finding that the truth-telling might be more difficult than the concealment lie (and closely aligned confidence & anxiety ratings). It's commonly stated that lying is more difficult than telling the truth, which is clearly not the case here, so perhaps a

short riff on this in the discussion is merited? [Stet. I see that you did touch on this in the discussion]. I certainly worry that similar patterns of results may be sitting in file drawers as a simple comparison of Lie v Truth would appear null. Good.

Page 12, Line 31 – Deception judgements were recorded using likert scale responses to “to what degree the interviewee was telling the truth” page 9, line 8 and then you present X2 analyses. So you categorised responses on the 1-7 scale? A quick comment on how that categorisation was achieved? Just for my sanity.

Experiment 1 results:

Generally clear presentation. Thank you.

Page 14, line 3 - I appreciate the unpacking of each limb results at page 14, line 3. As I think I asked above, I was curious if the averaging across limbs might have been problematic, in that perhaps it illustrates gross postural shifts or something other than motor mimicry. Am I right in thinking that this suggests limb to limb specificity? And if so, did you mention the orientation of interviewer/interviewee? And is there any pattern between the hands? Ipsilateral (if memory serves on the terminology)...

Page 14, line 31 – As this is a within-subject manipulation of veracity, I find it unclear when you speak of liars and truth tellers, and would prefer discussion of condition specific differences. I'm having trouble now reconciling the results with the concealment lie and lowest level of mimicry, then truth showing increased mimicry. Does this fit with the alternative explanation? Maybe it does. A quick comment on this might help me.

Experiment 2

Page 15, line 3 - I like the attention manipulations and use of two control groups to monitor the effect of added attentional demands to an alternative channel. But again, this suggests that interviewees will be mimicking the interviewer, which entirely makes sense, but I feel like I had to do quite a bit of work to be confident in that not being silly. My suggestion would be to cover off this potential misunderstanding in the introduction.

Page 18, line 29. So the results persist, whereby more mimicry is seen under conditions of increasing cognitive load. Nice.

Page 19, line 44 – I tend to agree with the assertion that the 2nd experiment covers off the alternative explanation. Nicely played.

Page 19, line 53 – I was wondering if you'd touch on the IDT assertion that when lying, individuals might attend more to the target to monitor suspicion etc. I think that the current task isn't designed to fully test this assertion, and I'd worry that an individual, who is required to deliver at least 2 lies and a truth in random order, might adopt a 'deceptive' approach throughout – which would give the same pattern of results you present, no? I don't have a problem with the suggestion you present, but it slightly jumped off the page as unnecessary.

General Discussion

Page 20, line 29 - Is there a typo? “in the context OF decision-making” ??

Page 20, line 31 – I'd probably pop a pair of “ around 'System 1' and 'System 2'. But I'm not fully on top of the RS style guide.

Page 20, line 47. Attempts plural? Something catches in the first part of the line.

Page 21, line 6 – Is it 'to what extent liars can'...? And again, I urge care in the suggestion that monitoring of motor mimicry might lend itself to lie detection efforts without carefully highlighting that even the results presented here don't immediately discriminate honesty from deception in any clear way. The easy lie obviously spoils that party. Not to say that I disagree though!

Page 22, line 17 – I'm pleased to see the discussion of easy truth/difficult lie assumption – and data to confound this assumption! I was thinking that a concealment in this scenario would also be easier by dint of the fact that it is a lesser communicative task; there is no fabrication involved (as per design) and likely just a denial of 'anything odd having happened'... I do however believe that telling a lie about nothing odd having happened (if it had) would be more difficult than

telling the truth about nothing odd having happened (if it hadn't)... Obviously, this is me just speculating.

Final comments. Very thoughtful closure. Enjoyed this paper very much.

Reviewer: 2

Comments to the Author(s)

GENERAL

A general comment refers to the concept of mimicry. Could the authors please define it clearly and define their operationalization clearly – early on. This is crucial because mimicry is a form of interpersonal coordination, like social synchrony and the both are sometimes used interchangeably. But mimicry specifically implies imitation and copying. Therefore, an evaluation of whether the DTW is an appropriate method to quantify mimicry and whether an interpretation of findings in relation to “automatic imitation” is warranted, depends on the author’s definition. A general switch to a more general “coordination” terminology instead of “mimicry” would be in acceptable, otherwise, I would like to ask the authors to offer a definition and operationalization of mimicry in their study and justifications for the use of the DTW and the interpretation of results that would be consistent with their definition of mimicry.

Another more general comment refers to the need for a clearer and more detailed explanation of the use of gyroscope data with the DTW method.

Finally, I find that Experiment 2 has a design that does not manage 100% to manipulate attention and cognitive load in the way it was intended and I would like more clarification from the authors or a discussion of this as a limitation.

More comments and details follow below:

ABSTRACT

1. The abstract mentions Experiment 2 but Experiment 1 should be clearly named as well. DTW as a method should also be mentioned in the abstract

INTRODUCTION

2. I recommend a clear introduction of terminology of terms like “interviewer” and “interviewee = liar”

3. The link between lying as a stressor and freezing is not convincing and a reference for “and may subsequently lead to freezing” could help.

4. I suggest that the introduction finishes with a clear summary of the research gaps and therefore a phrasing of the research aim(s). The second to last paragraph in the introduction does this well and leads the reader to understand why cognitive load matters. But the final paragraph ends too abruptly.

5. The authors mention in the second to last paragraph that there are two opposing ways in which lying may affect mimicry. But it is unclear what opposing means. I have attempted a summary and if this is correct I suggest that more signposting is used to make this clearer in the text: Way 1: Lying is cognitively demanding hence will lead to increased mimicry. Way 2: Lying is stressful which can lead to freezing hence decreased mimicry. A good way to signpost would be to actually state the theory first before then deepening the rationale.

6. Because the second to last paragraph is so long, I recommend starting the last one with more signposting like “The other way in which lying may affect mimicry is related to...”

7. This sentence and reference are unclear “In humans, mimicry can facilitate information elicitation and elicit cues to deceit (Shaw et al., 2015), could the. Authors explain what they mean?”

8. The introduction should have a clear literature discussion on the link between mimicry and attention. The discussion of Exp 1 says “This positive association between mimicry and attention

has previously been observed in relation to facial mimicry (Likowski, Muhlberger, Seibt, Pauli, & Weyers, 2008).” But this would need to be elaborated and ideally in the introduction, because that sets up both studies in the main body

EXPERIMENT 1

9. Please explain very briefly the terminology of fabrication and lying early on as the terms are introduced, for example also reverse fabrication. This becomes evident further on, but may be confusing the first time they are mentioned to readers who are not versed in the deception literature

10. Please formulate clear hypotheses that can then be referred to in the discussion

11. “A minimum sample size of 15 participants per condition was established in a pilot study.” How was this established? Can the authors give more details regarding the power analysis?

12. What is the difference between the difficult and very difficult tasks? The reverse order becomes clear later in the text but should be clarified as early as possible when the authors describe the Pre-interview Tasks, under “Difficult and very difficult lie task”

13. What if participants were already familiar with the game Clue compared to completely naïve participants. Do the authors think this might be a source of variability, could this be controlled for?

14. Please explain what was the cover story used to explain to participants why they are being made to wear the sensors?

15. The authors briefly mention that because it was not possible to ensure that the devices began recording at the exact same moment, they aligned the data of each device. Please explain how this alignment was achieved.

16. How were missing data handled? For example, if head movement is missing in a pair but all other sensors aren't? I assume that when reporting the individual body parts, the sample size varies, but what about the average score?

17. Does DTW take into account only timing, or angle size as well? So, if the authors are looking at gyroscope data, what exactly are they using as raw data for the DTW analysis?

18. Can the authors give evidence of DTW being used in either mimicry or joint action literature and if yes with what kind of data? Please make it clear if this is something previously validated or something exploratory and new. In the latter case it would be good to have a justification for including this method that shows how this is advantageous by comparison to previous (e.g. cross-correlation, wavelet coherence, cross-recurrence) methods.

19. We averaged the four DTW scores per limb - unclear what are the four scores? Do you mean body part instead of limb? You had head, rib cage and 2 wrists. Correct this for the rest of results/discussion section. Limbs mean arms or legs only

RESULTS

20. Is there a correction for multiple comparisons applied when performing the pairwise comparisons?

DISCUSSION

21. Please see my general comment on mimicry vs coordination/synchrony - it matters what we call it. Because mimicry is imitation, whereas a general synchrony or coordination does not necessarily entail imitation at all. And this will affect how the authors interpret the results

22. Note that the way the introduction is set up, it leads the reader to expect a discussion of the idea around how stressful the lying is perceived/freezing. This should be discussed here and how this Exp 1 may be used to corroborate one or the other opposing theories.

EXPERIMENT 2

23. If liars are thought to monitor the partners more than truth tellers, this means they pay more attention to them. But is there any reason to think that this would be the case for nonverbal monitoring and not also general monitoring, including verbal monitoring? This is important because it influences the authors' hypotheses for Experiment 2 and it is not clear why they

hypothesize “If following instructions increases cognitive load in general, then mimicry levels in the nonverbal and verbal instruction condition will be higher than mimicry levels in the no instruction condition”). And then we have the factor related to attention (“nonvb”, “verbal” and “none”), and we’d expect a replication of findings in the “none” condition. But then, also, depending on the hypothesis, some interactions. I find it hard to map this design and hypotheses onto what has been actually done and I would like more clarifications.

24. “First, in order to manage the design complexity of the experiment, we removed the difficult lie condition.” I find that if Experiment 2 was done as a follow-up experiment, two task loads would have been enough, e.g. “easy lie and difficult fabrication” to make sure the choice is of those tasks that showed clearest difference. Why include or exclude anything else?

25. Although explained in different ways, it is unclear how many sample pairs there were per condition? Please clarify.

26. I find it difficult to believe that the instruction “pay extra attention to the nonverbal behaviour of the interviewer” is enough to create attention AND cognitive load. Also, the check in the questionnaire, is too vague “the amount of attention they paid to ‘the interviewer’s nonverbal behaviour’ (1) versus ‘the interviewer’s verbal behaviour’ (7).” Why not ask them to count something specific in their partner’s nonverbal behavior like the raising of the left arm, or ask them to estimate the speed of the movement. Otherwise, the claim “Using both control groups allowed us to separate the effect of attention to nonverbal behaviour from the effect of increased cognitive load caused by adding the additional task of having to follow the attention instructions.” is not 100% convincing

27. “This suggests, in contrast to previous assertions (Buller & Burgoon, 1996; Kassin & Gudjonsson, 2004; Schweitzer et al., 2002), that interviewees do not increasingly monitor their interaction partner when lying, or at least they are not consciously aware of doing so.” How did the measure in this study differ to those referenced studies? This study included very explicit questions to measure a very implicit phenomenon, I wonder if an implicit phenomenon may not be best captured by an implicit measure? Perhaps eye-tracking might be a good implicit measure to be used? Again, what did the authors in the referenced papers use?

GENERAL DISCUSSION

28. Please add a conclusion

OTHER

29. The supplementary material contains some redundant information with regard to the DTW (i.e. it is explained in the main manuscript text).

Author's Response to Decision Letter for (RSOS-200839.R0)

See Appendix A.

RSOS-200839.R1 (Revision)

Review form: Reviewer 2

Is the manuscript scientifically sound in its present form?

Yes

Are the interpretations and conclusions justified by the results?

Yes

Is the language acceptable?

Yes

Do you have any ethical concerns with this paper?

No

Have you any concerns about statistical analyses in this paper?

No

Recommendation?

Accept as is

Comments to the Author(s)

I appreciate the care with which the authors have ensured that the work is thoroughly communicated (e.g. added information regarding the DTW) and the open materials. Also, upon re-reading, I stand corrected on my 26th point: I was skeptical on the claim "Using both control groups allowed us to separate the effect of attention to nonverbal behaviour from the effect of increased cognitive load caused by adding the additional task of having to follow the attention instructions" - but I had misunderstood it originally. All my comments have been thoroughly addressed.

Decision letter (RSOS-200839.R1)

Dear Dr Taylor,

It is a pleasure to accept your manuscript entitled "A Liar and a Copycat: Nonverbal Coordination Increases with Lie Difficulty" in its current form for publication in Royal Society Open Science. The comments of the reviewer(s) who reviewed your manuscript are included at the foot of this letter.

on behalf of Dr Carina de Klerk (Associate Editor) and Essi Viding (Subject Editor)
openscience@royalsociety.org

Associate Editor Comments to Author (Dr Carina de Klerk):

Dear authors,

Thank you for submitting your revised manuscript and for addressing all the comments raised by the reviewers. Your paper has now been accepted for publication.

Best wishes,
Carina de Klerk

Reviewer comments to Author:

Reviewer: 2

Comments to the Author(s)

I appreciate the care with which the authors have ensured that the work is thoroughly communicated (e.g. added information regarding the DTW) and the open materials. Also, upon re-reading, I stand corrected on my 26th point: I was skeptical on the claim "Using both control groups allowed us to separate the effect of attention to nonverbal behaviour from the effect of increased cognitive load caused by adding the additional task of having to follow the attention instructions" - but I had misunderstood it originally. All my comments have been thoroughly addressed.

Appendix A

Dear Carina and Reviewers

Many thanks for such helpful reviews. It is very kind of you to provide such thoughtful reviews at this time, with all the pressures the pandemic has brought. (Reviewer 1 you certainly were not a dreaded Reviewer 2! And, to be clear, nor was the actual Reviewer 2).

We've addressed every comment and describe how we did so below. If you feel we missed something, or have further suggestions, please do let us know.

Page 3, Line 49 –It has been a while since I read Shaw et al., 2015 and query whether the claims of increased information elicitation and deceit cue elicitation are entirely justified based on the results of that paper. My recall is that the results are not quite that clear-cut. And in that paper, if memory serves, it is interviewers who mimic the interviewees.

Yes, you're right. This was a sloppy account of the research by us. We've now edited the description to give an accurate account.

Page 6, Line 10 - Respectable sample size on the face of it. I'd be interested to see more of the information on sample size estimate (such as effect size achieved in the Pilot study mentioned – page 6, line 29).

We now give an account of the pilot work we did to determine the sample size.

Page 6, Line 45 – Initially unclear why the design is mixed with truth/concealment (WS) and the difficult / very difficult lie (BS). A quick comment on this would help. Even by page 9, line 29 this is somewhat unclear as to why this would be designed as such. Quick comment.

We now give what is hopefully a useful two lines outlining the study design at the top of the Procedure section.

Page 7, Line 31 – Confederate task query

We suspect the Reviewer forgot to elaborate on this comment and I'm afraid despite our best efforts we couldn't infer what was likely meant. We're happy to address it in a revision.

Page 7, Line 54 – I like the design of the confederate task in the difficult lie condition. It achieves the blend of a clear task, with achievable outcome, but with sufficient variability/detail and clearly an aspect of motivation by incentive. I often work with students and collaborators who are keen to develop deception tasks of this type, with a view to operationalising deception with the goal of eliciting certain types of behaviour. If space permits, I'd love to see a couple of points from the authors on the key features of the task that illustrate the thought put into the development of the task. I think it would be useful for other researchers interested in this type of experimental development.

We now give a brief account of why we liked the Cluedo task. We value open science very highly, and have uploaded all materials to the open online repository, including puzzle and Cluedo information and a detailed experimenter script. Other researchers are very welcome to use these materials for replication efforts or their own experiments.

Page 11, Line 22 – It occurs to me at this point that no mention of 'direction' of mimicry has been suggested up front explicitly, although since the would-be liar is the participant under increased cognitive load, and under such conditions mimicry appears to increase, it might be fair to assume the naïve assumption that the liar mimics the interviewer, or am I wrong? The symmetry decision within DTW therefore seems to require some support – or is this a more liberal approach than assuming Interviewer behavioural cues preceding the interviewees mimicry.... Or maybe the specific nature/direction of the mimicry (or any debate around it) is worth highlighting in the introduction perhaps? I'm not fully up to speed with the mimicry literature, so it may be that distinctions around sequence or 'direction' are ignorant, but I think

a naïve reader would welcome information on this. [I do see that you deal with some of these points in the General Discussion – nicely]. I still wonder if this could be pre-empted somewhat. Our use of symmetry DTW reflected our desire to look at coordination as a whole an unease about considering mimicry as a solo process, particularly given our opening argument. We thus felt that a symmetric measure made the least number of assumptions about the data and did not artificially curtail the measure. For example, if the person ‘leading’ coordination oscillates between interviewer and interviewee, not allowing DTW to adjust the two time series to capture that may have the knock on effect that the subsequent interviewee led coordination is also not captured, as it would then fall outside of the window of alignment.

Page 11, line 31 – I am aware of the expertise of the research group with regards to the analysis of NVB, so please bear with me – these may be silly questions. I am keen to understand what the DTW scores will actually be telling me and at what level of granularity. Synchrony? Magnitude? Is there any opportunity to explore the patterns in the various limbs. I’m interested to know if there is spatial matching etc.

We have re-written the second paragraph of p. 11-12 to provide more clarity over what the DTW score is showing. We also clarify here that we will examine both individual limb and an overall measure, to prelude the analyses that appear on p. 13.

Page 12, Line 3 – I am not surprised by the finding that the truth-telling might be more difficult than the concealment lie (and closely aligned confidence & anxiety ratings). It’s commonly stated that lying is more difficult than telling the truth, which is clearly not the case here, so perhaps a short riff on this in the discussion is merited? [Stet. I see that you did touch on this in the discussion]. I certainly worry that similar patterns of results may be sitting in file drawers as a simple comparison of Lie v Truth would appear null. Good.

We fully agree. We also believe it is important to spend a bit more attention to this finding. It is now mentioned on p. 13 and critically discussed on p. 24 and 25 to reinforce this as a take away message.

Page 12, Line 31 – Deception judgements were recorded using likert scale responses to “to what degree the interviewee was telling the truth” page 9, line 8 and then you present X2 analyses. So you categorised responses on the 1-7 scale? A quick comment on how that categorisation was achieved? Just for my sanity.

Thanks for spotting this. We now explain in more detail how we computed correct / incorrect scores based on the Likert scale responses on p. 10 (methods) and p. 14 (results). The chi square test should make sense now.

Page 14, line 3 - I appreciate the unpacking of each limb results at page 14, line 3. As I think I asked above, I was curious if the averaging across limbs might have been problematic, in that perhaps it illustrates gross postural shifts or something other than motor mimicry. Am I right in thinking that this suggests limb to limb specificity? And if so, did you mention the orientation of interviewer/interviewee? And is there any pattern between the hands? Ipsilateral (if memory serves on the terminology)...

p. 9 now describes how the participants were positioned during the interview. p. 15 now presents the results for both ipsilateral and isolateral coordination results.

Page 14, line 31 – As this is a within-subject manipulation of veracity, I find it unclear when you speak of liars and truth tellers, and would prefer discussion of condition specific differences. I’m having trouble now reconciling the results with the concealment lie and lowest level of mimicry, then truth showing increased mimicry. Does this fit with the alternative explanation? Maybe it does. A quick comment on this might help me.

We have now rewritten the Discussion so that it discusses the results of Study 1 by condition, which should help clarify the argument for looking at attention.

Experiment 2

Page 15, line 3 - I like the attention manipulations and use of two control groups to monitor the

effect of added attentional demands to an alternative channel. But again, this suggests that interviewees will be mimicking the interviewer, which entirely makes sense, but I feel like I had to do quite a bit of work to be confident in that not being silly. My suggestion would be to cover off this potential misunderstanding in the introduction.

As above we do now address this in more detail.

Page 18, line 29. So the results persist, whereby more mimicry is seen under conditions of increasing cognitive load. Nice.

Yes!

Page 19, line 44 – I tend to agree with the assertion that the 2nd experiment covers off the alternative explanation. Nicely played.

Thanks. There are of course always possible alternatives, but we agree it does cover it off to a large extent.

Page 19, line 53 – I was wondering if you'd touch on the IDT assertion that when lying, individuals might attend more to the target to monitor suspicion etc. I think that the current task isn't designed to fully test this assertion, and I'd worry that an individual, who is required to deliver at least 2 lies and a truth in random order, might adopt a 'deceptive' approach throughout – which would give the same pattern of results you present, no? I don't have a problem with the suggestion you present, but it slightly jumped off the page as unnecessary.

We have been careful with our language here, trying simply to say the evidence doesn't fit IDT. We have also added a sentence clarifying that this is not a direct test and other explanations are possible.

General Discussion

Page 20, line 29 - Is there a typo? "in the context OF decision-making" ??

Yes well spotted. Now corrected.

Page 20, line 31 – I'd probably pop a pair of ' around 'System 1' and 'System 2'. But I'm not fully on top of the RS style guide.

Done.

Page 20, line 47. Attempts plural? Something catches in the first part of the line.

Indeed. Done.

Page 21, line 6 – Is it 'to what extent liars can'...? And again, I urge care in the suggestion that monitoring of motor mimicry might lend itself to lie detection efforts without carefully highlighting that even the results presented here don't immediately discriminate honesty from deception in any clear way. The easy lie obviously spoils that party. Not to say that I disagree though!

We've corrected the typo—thanks—and added a sentence to remind the reader about the fact difficulty rather than deception was the driver of our finding.

Page 22, line 17 – I'm pleased to see the discussion of easy truth/difficult lie assumption – and data to confound this assumption! I was thinking that a concealment in this scenario would also be easier by dint of the fact that it is a lesser communicative task; there is no fabrication involved (as per design) and likely just a denial of 'anything odd having happened'... I do however believe that telling a lie about nothing odd having happened (if it had) would be more difficult than telling the truth about nothing odd having happened (if it hadn't)... Obviously, this is me just speculating.

As above, we've added to the paragraph discussing this issue in the Discussion, to highlight the need for careful research design.

Final comments. Very thoughtful closure. Enjoyed this paper very much.

Thank you. And, thank you for your helpful comments.

Reviewer: 2

Comments to the Author(s)

GENERAL

A general comment refers to the concept of mimicry. Could the authors please define it clearly and define their operationalization clearly – early on. This is crucial because mimicry is a form of interpersonal coordination, like social synchrony and the both are sometimes used interchangeably. But mimicry specifically implies imitation and copying. Therefore, an evaluation of whether the DTW is an appropriate method to quantify mimicry and whether an interpretation of findings in relation to “automatic imitation” is warranted, depends on the author’s definition. A general switch to a more general “coordination” terminology instead of “mimicry” would be in acceptable, otherwise, I would like to ask the authors to offer a definition and operationalization of mimicry in their study and justifications for the use of the DTW and the interpretation of results that would be consistent with their definition of mimicry.

Thank you for this excellent point. Given our measure and the definitions associated with mimicry and coordination, it is clear our research speaks to nonverbal coordination. We have thus changed the manuscript throughout to recognise this, including defining the difference in the introduction, describing in detail how the DTW measure afford us a measure of coordination, and discussing the Results in relation to coordination and not imitation.

Another more general comment refers to the need for a clearer and more detaile explanation of the use of gyroscope data with the DTW method.

We now give a more detailed account of the DTW method.

Finally, I find that Experiment 2 has a design that does not manage 100% to manipulate attention and cognitive load in the way it was intended and I would like more clarification from the authors or a discussion of this as a limitation.

We give a more detailed account of the Method (please see below) and believe it does afford us sufficient evidence to rule out the attention alternative, though we welcome further challenge from the Reviewer. Also, all materials including attention instructions and the experimenter script are uploaded to the open online repository to facilitate understanding, replication, and designing deception experiments by other researchers.

ABSTRACT

1. The abstract mentions Experiment 2 but Experiment 1 should be clearly named as well. DTW as a method should also be mentioned in the abstract

We reworded the Abstract so that both experiments are now explicitly mentioned.

INTRODUCTION

2. I recommend a clear introduction of terminology of terms like “interviewer” and “interviewee = liar”

Thank you for pointing this out. We have re-written the first couple of sentences of the opening to make clear our terminology.

3. The link between lying as a stressor and freezing is not convincing and a reference for “and may subsequently lead to freezing” could help.

We now give an explicit reference for the ‘freezing’ suggestion.

4. I suggest that the introduction finishes with a clear summary of the research gaps and therefore a phrasing of the research aim(s). The second to last paragraph in the introduction does this well and leads the reader to understand why cognitive load matters. But the final paragraph ends too abruptly.

We have re-written the ending of the last paragraph to make clearer the alternative hypothesis. Combined with our response to comment #5, we believe this makes a much clearer articulation of the paper's predictions.

5. The authors mention in the second to last paragraph that there are two opposing ways in which lying may affect mimicry. But it is unclear what opposing means. I have attempted a summary and if this is correct I suggest that more signposting is used to make this clearer in the text: Way 1: Lying is cognitively demanding hence will lead to increased mimicry. Way 2: Lying is stressful which can lead to freezing hence decreased mimicry. A good way to signpost would be to actually state the theory first before then deepening the rationale.

Yes this account is correct. We now signpost with an additional sentence at the top of each paragraph.

6. Because the second to last paragraph is so long, I recommend starting the last one with more signposting like "The other way in which lying may affect mimicry is related to..."

Done. Thanks for this suggestion—it is certainly clearer now.

7. This sentence and reference are unclear "In humans, mimicry can facilitate information elicitation and elicit cues to deceit (Shaw et al., 2015), could the. Authors explain what they mean?"

This point was also correctly picked up by Reviewer 1. We have now given a more accurate account of the study.

8. The introduction should have a clear literature discussion on the link between mimicry and attention. The discussion of Exp 1 says "This positive association between mimicry and attention has previously been observed in relation to facial mimicry (Likowski, Muhlberger, Seibt, Pauli, & Weyers, 2008)." But this would need to be elaborated and ideally in the introduction, because that sets up both studies in the main body

We now give a more comprehensive account of the potential link between attention and coordination on pp. 16-18. We do so in the introduction of Exp. 2, rather than at the beginning, as we feel this gives a more genuine account of the research and avoids confusion over the purpose of Exp 1.

EXPERIMENT 1

9. Please explain very briefly the terminology of fabrication and lying early on as the terms are introduced, for example also reverse fabrication. This becomes evident further on, but may be confusing the first time they are mentioned to readers who are not versed in the deception literature

We have added explanation doing this.

10. Please formulate clear hypotheses that can then be referred to in the discussion

We have added a sentence doing this.

11. "A minimum sample size of 15 participants per condition was established in a pilot study." How was this established? Can the authors give more details regarding the power analysis?

We now give details of the work that informed our sample size.

12. What is the difference between the difficult and very difficult tasks? The reverse order becomes clear later in the text but should be clarified as early as possible when the authors describe the Pre-interview Tasks, under "Difficult and very difficult lie task"

Thank you for pointing this out. We now give a description at the beginning of the ‘Difficult and very difficult lie task’ section.

13. What if participants were already familiar with the game Clue compared to completely naïve participants. Do the authors think this might be a source of variability, could this be controlled for?

One of our reasons for choosing Clue/Cluedo is that its components are familiar to crime stories more generally, and so those elements will have some level of familiarity for participants even if they haven’t played the game. Of course, this does not rule out the familiarity effect suggested here, which is not something we controlled for. Fortunately, such variation would increase the within condition variance and make it harder for us to find evidence to reject the null hypothesis. Moreover, such variation is true for liars who often vary in the extent to which they have underlying familiarity with what they are lying about, so it holds a degree of ecological validity too.

14. Please explain what was the cover story used to explain to participants why they are being made to wear the sensors?

No cover story was used when putting the WiTilts onto participants. Instead we relied on our debrief question where we looked for evidence that participants knew mimicry was the focus of the study. As reported on p.11, none did.

15. The authors briefly mention that because it was not possible to ensure that the devices began recording at the exact same moment, they aligned the data of each device. Please explain how this alignment was achieved.

Now explained on p. 11. Each data point had a corresponding time stamp and so we aligned the streams by matching them up to the first timestamp common to both recordings.

16. How were missing data handled? For example, if head movement is missing in a pair but all other sensors aren’t? I assume that when reporting the individual body parts, the sample size varies, but what about the average score?

Where a DTW score for one of the limbs was missing, we used the average of the available scores. This is now described on p. 13 (last sentence before the results section).

17. Does DTW take into account only timing, or angle size as well? So, if the authors are looking at gyroscope data, what exactly are they using as raw data for the DTW analysis?

The gyro data gives us the size but not direction of body part movement at each time point. We now clarify this and give an example on p.12. We chose the gyro over accelerometer data as the single measure is easier to understand and, with only 4DOF, we had no way of converting the accelerometer data into an accurate mimicry score. As we describe in the Discussion, using a 6DOF systems would allow matching of timing, angle, and magnitude, and this would be a useful direction for future research.

18. Can the authors give evidence of DTW being used in either mimicry or joint action literature and if yes with what kind of data? Please make it clear if this is something previously validated or something exploratory and new. In the latter case it would be good to have a justification for including this method that shows how this is advantageous by comparison to previous (e.g. cross-correlation, wavelet coherence, cross-recurrence) methods.

We are not aware of other research using the DTW method in the mimicry or joint action literature. We have now clarified this on p.11 and added a description of why we believe this is a more flexible approach than previously used methods.

19. We averaged the four DTW scores per limb - unclear what are the four scores? Do you mean body part instead of limb? You had head, rib cage and 2 wrists. Correct this for the rest of results/discussion section. Limbs mean arms or legs only

We do, and you are absolutely correct re terminology. Thank you for picking this up. We have now replaced limb with body part throughout.

RESULTS

20. Is there a correction for multiple comparisons applied when performing the pairwise comparisons?

We use a Bonferroni correction to aid interpretation of the planned pairwise comparison. This is now made explicit in the description for Study 1 (p. 15) and Study 2 (p. 19 & 20).

DISCUSSION

21. Please see my general comment on mimicry vs coordination/synchrony - it matters what we call it. Because mimicry is imitation, whereas a general synchrony or coordination does not necessarily entail imitation at all. And this will affect how the authors interpret the results.

Addressed above.

22. Note that the way the introduction is set up, it leads the reader to expect a discussion of the idea around how stressful the lying is perceived/freezing. This should be discussed here and how this Exp 1 may be used to corroborate one or the other opposing theories.

We have expanded the first paragraph of the Discussion to make explicit the original comparison and the implications of the data.

EXPERIMENT 2

23. If liars are thought to monitor the partners more than truth tellers, this means they pay more attention to them. But is there any reason to think that this would be the case for nonverbal monitoring and not also general monitoring, including verbal monitoring? This is important because it influences the authors' hypotheses for Experiment 2 and it is not clear why they hypothesize "If following instructions increases cognitive load in general, then mimicry levels in the nonverbal and verbal instruction condition will be higher than mimicry levels in the no instruction condition". And then we have the factor related to attention ("nonvb", "verbal" and "none"), and we'd expect a replication of findings in the "none" condition. But then, also, depending on the hypothesis, some interactions. I find it hard to map this design and hypotheses onto what has been actually done and I would like more clarifications.

This is one of the reasons we included both the verbal and nonverbal attention conditions. Our view is that a 'general attention' interpretation of the instruction would lead to similar findings in the verbal instruction and nonverbal instruction condition. By contrast, support for H1 came from finding no effects of the attention manipulation. The various 'possibilities' and how they relate to our hypothesis is now clarified on p. 16.

24. "First, in order to manage the design complexity of the experiment, we removed the difficult lie condition." I find that if Experiment 2 was done as a follow-up experiment, two task loads would have been enough, e.g. "easy lie and difficult fabrication" to make sure the choice is of those tasks that showed clearest difference. Why include or exclude anything else?

It is true that two conditions would have possibly sufficed, but we didn't want to make the assumption that veracity had no role to play in the results. It felt appropriate, and more akin to a replication, if we retained both difficulty and veracity within the Experiment. We now clarify that on p. 18.

25. Although explained in different ways, it is unclear how many sample pairs there were per condition? Please clarify.

Done, on p. 18.

26. I find it difficult to believe that the instruction "pay extra attention to the nonverbal behaviour of the interviewer" is enough to create attention AND cognitive load. Also, the check in the questionnaire, is too vague "the amount of attention they paid to 'the interviewer's nonverbal behaviour' (1) versus 'the interviewer's verbal behaviour' (7)." Why not ask them to count something specific in their partner's nonverbal behavior like the raising of the left arm, or ask them to estimate the speed of the movement. Otherwise, the claim "Using both control

groups allowed us to separate the effect of attention to nonverbal behaviour from the effect of increased cognitive load caused by adding the additional task of having to follow the attention instructions.” is not 100% convincing

We respectfully agree with Reviewer 1 and believe the manipulation does offer a convincing test of the attention manipulation. The attention instruction sought to increase attention, not attention AND cognitive load. We used the verbal behaviour instruction to provide a comparison point just in case the attention condition did affect load. That is, we wanted to know if any observed differences were the result of paying attention per se or paying attention to nonverbal behaviour; nonverbal being the behaviour of interest. A manipulation such as ‘counting raising of the left arm’ would have been possible, but we felt it would overburden the participant. Moreover, it also does not have any ecological basis, unlike attending to the interviewer’s behaviour, which is reported by liars as something they do. Finally, for a counting manipulation to provide a better measure than our Likert scale of what participants did, we would need to film and count the amount of arm movement. Otherwise, participants could simply make up the number.

Nonetheless, we highlight this limitation of the manipulation during the Discussion on p.22.

27. “This suggests, in contrast to previous assertions (Buller & Burgoon, 1996; Kassin & Gudjonsson, 2004; Schweitzer et al., 2002), that interviewees do not increasingly monitor their interaction partner when lying, or at least they are not consciously aware of doing so.” How did the measure in this study differ to those referenced studies? This study included very explicit questions to measure a very implicit phenomenon, I wonder if an implicit phenomenon may not be best captured by an implicit measure? Perhaps eye-tracking might be a good implicit measure to be used? Again, what did the authors in the referenced papers use?

Our suggestion here related to a theoretical account of how deception works within interaction. We have now clarified this in the paragraph. We do agree that eye tracking is a good way to measure what people pay attention too and have mentioned it as a suggestion for future research projects.

GENERAL DISCUSSION

28. Please add a conclusion

We have added a conclusion.

OTHER

29. The supplementary material contains some redundant information with regard to the DTW (i.e. it is explained in the main manuscript text).

We have removed the duplication.